# Healthy or Unhealthy? The Cocktail of Health-Related Behavior Profiles in Spanish Adolescents

**DOI:** 10.3390/ijerph16173151

**Published:** 2019-08-29

**Authors:** Javier Sevil-Serrano, Alberto Aibar-Solana, Ángel Abós, José Antonio Julián, Luis García-González

**Affiliations:** 1Faculty of Health and Sport Sciences, Department of Didactics of the Musical, Plastic and Corporal Expression, University of Zaragoza, 22001 Huesca, Spain; 2Faculty of Social Sciences and Humanities, Department of Didactics of the Musical, Plastic and Corporal Expression, University of Zaragoza, 22003 Huesca, Spain

**Keywords:** physical activity, health behavior, sedentary time, diet, sleep, screen time, cluster, sex, body mass index

## Abstract

The aim of this study was to identify the prevalence and clustering of health-related behaviors in Spanish adolescents and to examine their association with sex, body mass index (BMI), different types of sedentary screen time, and adherence to 24-hour movement guidelines. A final sample of 173 students (M = 12.99 ± 0.51) participated in this study. Cluster analysis was conducted based on five health-related behaviors: PA and sedentary time derived from accelerometers, as well as healthy diet, sedentary screen time, and sleep duration derived from self-reported scales. Recommendations for 24-hour movement guidelines (i.e., physical activity (PA), screen time, and sleep duration) were analyzed both independently and combined. A total of 8.9% of the sample did not meet any of the guidelines, whereas 72.3%, 17.3%, and 1.7% of the sample met 1, 2, or all 3 guidelines, respectively. Six distinct profiles were identified, most of them showing the co-occurrence of healthy- and unhealthy-related behaviors. Given that most of the adolescents failed to meet the combination of PA, screen time, and sleep duration guidelines, these findings suggest the necessity to implement school-based interventions that target multiple health behaviors, especially because (un)healthy behaviors do not always cluster in the same direction.

## 1. Introduction

High levels of physical activity (PA) [1], low levels of sedentary screen time (ST) (particularly from television, computer, and video games) [2], high quality diet [3], and sufficient sleep duration [4], are independently associated in adolescents with a wide range of physical, social, and mental health benefits. However, a simultaneous positive combination of these health-related behaviors could even maximize those health benefits among adolescents [5].

New PA, sedentary screen time, and sleep duration guidelines have been recently proposed according to the integrative approach of the 24-hour time use continuum [6]. While a small proportion of adolescents met PA, sedentary screen time, and sleep duration recommendations separately, the proportion of those meeting all three recommendations seems to be even lower [7]. One study, conducted in 12 countries, reported that approximately just 7% of children of the total sample met all three recommendations (i.e., PA, screen time, and sleep duration), ranging from 2% compliance in China to 15% in Australia [7]. Compliance with health behavior recommendations may be affected by social, cultural or economic factors across countries [8]. Therefore, future research in adolescents from different countries is warranted to further understand these health-related behaviors.

Independent associations of PA, ST, sleep duration, and healthy diet with health indicators in youth have usually been analyzed following a variable-centered approach [5]. However, different combinations of these health-related behaviors can co-occur in individuals at the same moment, facilitating the simultaneous existence of both health-promoting and health-risk behaviors [9,10]. Clustering PA, ST, and diet in children and adolescents is well-documented in research literature [9]. For example, the clustering pattern of PA, ST, diet quality, and sedentary screen time among older children, younger adolescents, and older adolescents has been recently identified [11]. Four different profiles were identified in the three mentioned groups: (a) a healthy lifestyle cluster (high MVPA, healthy diet, low sedentary screen time and low total ST), (b) a sedentary and healthy diet cluster, (c) a sedentary screen time cluster, and (d) a low MVPA and unhealthy diet cluster. However, interest in other health-related behaviors has recently increased in youth, such as in sleep duration due to its health effects and its interaction with other health-related behaviors [12] and adverse health outcomes such as overweight and obesity [13,14].

A previous systematic review conducted on adolescents showed that short sleep duration (i.e., <8 h/day in adolescents) [6] was positively associated with excessive ST and unhealthy diets [13]. However, no associations were found between short sleep duration and other health-related behaviors such as PA and sedentary screen time in the last-mentioned systematic review [13]. However, a recent systematic review and meta-analysis, conducted among adolescents, aged between 11 and 20 years old, reported a positive association between sedentary screen time and short sleep duration [15]. Although the relationship between long sleep duration and health-related behaviors has not been studied in depth, it seems that longer sleepers are more likely to report less screen time [8] and low PA levels [16]. Clustering sleep duration with other health-related behaviors may therefore contribute to a better understanding of health-related behavioral profiles in adolescents [17,18,19].

The clustering pattern of PA, ST, sleep duration, diet quality, and sedentary screen time among children from 12 countries has been recently identified [20,21]. The retained profiles among children were characterized by: (a) mostly healthy behaviors (e.g., high PA, low sedentary screen time, healthy diet, and moderate/high ST), (b) mostly risk behaviors (e.g., low PA, moderate sedentary screen time, unhealthy diet, and high ST), and (c) co-occurrence of healthy and unhealthy behaviors (e.g., high PA, high sedentary screen time, high ST, and healthy, and unhealthy diet). In all these studies no differences in sleep duration were found across profiles that could be explained by parents’ influence on the management of their children’s bedtime.

Transition from childhood to adolescence is associated with changes in multiple health-related behaviors [22]. Biological, environmental or social changes (e.g., puberty, new school environment), a higher level of autonomy to take decisions, cognitive development (e.g., maturing), or greater accessibility to electronic devices could explain these health-related behavioral changes [23,24]. Therefore, clustering those health-related behaviors (i.e., PA, ST, sleep duration, diet quality, and sedentary screen time) in adolescents is fully warranted.

Despite recommendations that suggest measuring PA and ST with accelerometers [25], most prior studies using cluster analysis have used self-reported measures of PA and ST [9]. Moreover, given that ST and sedentary screen time are associated differently with different health indicators [2,26], their clustering should be independently studied. To date, studies using cluster analysis have exclusively focused on two or three types of sedentary screen time (i.e., TV, computer and/or video games). However, a recent drastic increase in the use of mobile phone technologies by adolescents in developed countries [27], and health risks associated with their increased use (e.g., anxiety, stress, sleep disturbance, etc.) [28], may suggest the need to explore its prevalence and possible influence on clustering other health-related behaviors.

Clustering health-related behaviors may differ by sex and weight status [9]. Profiles characterized by high PA have been associated with boys, whereas profiles characterized by low PA have usually been associated with girls [9]. Regarding the association between weight status and health-related profiles, inconsistent results have been found. While most of the studies show mainly no associations, some studies have reported positive or negative associations [9]. Finally, given the differences in prevalence of screen-based sedentary behaviors and in compliance with the 24-hour movement guidelines among adolescents [7,29], the independent association of each type of sedentary screen time (i.e., TV, computer, video games, and mobile phone) and adherence to 1, 2, or all 3 recommendations for 24-hour movement guidelines with resulting profiles should also be studied in depth.

This study had two main objectives: (a) identify the prevalence and clustering of PA, ST, healthy diet, sedentary screen time, and sleep duration in adolescents; and (b) examine the association between the retained cluster solutions and sex, body mass index (BMI), different types of sedentary screen time (i.e., TV, computer, video games, and mobile phone), and adherence to the 24-hour movement recommendations (i.e., PA, screen time, and sleep duration).

## 2. Materials and Methods

### 2.1. Participants and Procedures

An initial convenience sample of 225 Spanish secondary school students were asked to participate (52.9 % girls; M = 13.06 ± 0.61). Written informed consent was obtained from both parents and adolescents. Overall, 210 participants completed the questionnaire and wore an accelerometer (93.33% response rate). Twenty-six students were excluded from the data set due to: 1) missing values, and 2) not meeting the accelerometer inclusion criteria (*n* = 184; 87.6% valid rate). In addition, 7 univariate outliers and 4 multivariate outliers were removed, resulting in a final sample of 173 students (54.3% girls; M = 12.99 ± 0.51). The Ethics Committee for Clinical Research of Aragon (CEICA) approved all procedures (PI15/0283).

### 2.2. Measures

#### 2.2.1. Sociodemographic Characteristics

Age, ethnicity, sex, weight, height, and socioeconomic status (SES) were self-reported. Age- and sex-specific BMI were calculated using the World Health Organization growth reference for children and adolescents [30]. Adolescents’ SES was measured using the self-reported Family Affluence Scale II (FAS II) [31].

#### 2.2.2. Physical Activity and Sedentary Time

Daily average moderate to vigorous physical activity (MVPA) and ST were objectively measured using the triaxial Actigraph GT3X accelerometer over a 7-day period. The epoch length was set at 15 s. Evenson cut-points were used to determine the daily time spent on MVPA and ST [32]. The inclusion criteria for valid accelerometry data were as follows: wearing the accelerometer for at least 4 valid days, including at least one weekend day [33], and a valid day was defined as at least 10 h and 8 h of wear time on weekdays and on weekend days, respectively [34]. Based on PA guidelines, students were classified into two groups: active (≥60 min/day) and inactive (<60 min/day) [6].

#### 2.2.3. Dietary Habits

Healthy diet consumption was assessed using the Spanish validated version of the Health Behavior in School Children (HBSC) questionnaire [35]. Students reported their weekly frequency of consumption of six types of healthy food items (e.g., fruit, vegetables, fish, etc.) in a 3-point scale from never to every day. A healthy diet index (0–6 range) was calculated by adding up all the responses and recoding the variable [35].

#### 2.2.4. Sedentary Screen Time

Daily average screen time of different sedentary screen time behaviors (i.e., TV, computer, video games, and mobile phone) was self-reported independently for school days and weekend days using four of the twelve sedentary behaviors of the Spanish version of Youth Leisure-time Sedentary Behavior Questionnaire (YLSBQ) [36]. Test-retest reliability of this instrument showed moderate to good agreement over a one-week interval. A version of this YLSBQ has also shown a moderate level of validity. The daily screen time of each behavior was calculated by adding up the average daily screen time at a ratio of 5:2. The weekday and weekend day screen time was calculated by adding up the time spent on these four behaviors on weekdays and weekend days, respectively. The total daily sedentary screen time was calculated by adding up the daily screen time of the four behaviors. Based on screen time guidelines, students were classified into two categories: meeting screen time guidelines (<2 h/day) and not meeting screen time guidelines (≥2 h/day) [6] for each screen time behavior and for the total daily sedentary screen time.

#### 2.2.5. Sleep Duration

The average daily sleep duration was self-reported independently for school and non-school days using a validated question adapted from the Spanish version of Pittsburgh Sleep Quality Index [37]. Firstly, the sleep period between students’ self-reported bedtime and wake time was calculated for school and non-school days. Secondly, the daily sleep duration was also calculated at a ratio of 5:2. Sleep period has shown moderate to strong level of validity with accelerometer data and adequate reliability [38]. Based on sleep duration guidelines in adolescents, students were classified into two groups: short/long sleepers (<8 h/day or >10 h/day) and normal sleepers (8–10 h) [6].

### 2.3. Data Analysis

Sociodemographic characteristics, prevalence of health-related behaviors, and compliance with specific and general combinations of the 24-hour movement guidelines were analyzed using descriptive statistics. Based on 24-hour movement guidelines for adolescents (i.e., ≥60 daily minutes of MVPA, <2 daily hours of sedentary screen time and 8 to 10 daily hours of sleep duration), students were categorized into two groups for each behaviour: meeting guidelines and not meeting guidelines [6]. A cluster analysis based on MVPA, ST, healthy diet, sedentary screen time, and sleep duration, was conducted to identify health profiles. Health-related behaviors were previously standardized (z-scores), and univariate and multivariate outliers were removed. A combination of hierarchical and non-hierarchical methods of cluster analysis were used following a two-step approach [39]. First, a hierarchical cluster analysis was conducted using Ward’s method to identify initial cluster centers. To identify the number of cluster solutions, the percentage of variance, and dendrogram were visually inspected [40]. Two to seven cluster solutions were performed. Second, the resulting centroids for each possible number of cluster solutions were used as non-random initial cluster centers in an iterative, non-hierarchical k-means clustering procedure [41]. The reliability and stability of the final cluster solution were examined by a double-split cross-validation method [42]. Analyses of cluster differences related to BMI and the four screen time behaviors were conducted using univariate and multivariate analysis of variance, respectively. Bonferroni’s post hoc tests were also conducted. Effect sizes above 0.01 were considered small, above 0.06 moderate, and above 0.14 large [43]. Finally, chi-square and Cramer’s V tests were used to determine the association of cluster solutions with sex and compliance with the 24-hour movement recommendations. Cramer’s V values above 0.10 were considered small, above 0.30 medium, and above 0.50 large [43]. Excitatory (>2) and inhibitory (<–2) relationships were analyzed using adjusted residuals. The criterion of significance was set at *p* < 0.05. All analyses were performed using SPSS Version 21.0.

## 3. Results

The final sample did not show statistically significant differences between sex and BMI (Table 1). A total of 8.9% of the sample did not meet any of the guidelines, whereas 72.3%, 17.3%, and 1.7% of the sample met 1, 2 or all 3 guidelines, respectively. Most of the students met sleep duration recommendations (89%), whereas a small proportion of students met PA (21.4%) and screen time recommendations (1.7%) (Table 2).

A final six-cluster solution emerged from the two-step approach analysis (Figure 1), explaining about 50% of each health-related behavior variance. This six-cluster solution showed good stability and replicability (K = 0.83). Health-related behaviors showed statistically significant differences across the six clusters (Table 3). The final cluster profiles can be essentially described as (1) Inactive Unhealthy Eaters: lowest PA and healthy diet; (2) Non-Technological Sitters: highest ST and lowest sedentary screen time, (3) Active: higher PA and lowest ST; (4) Technological Sleepyheads: highest sedentary screen time and sleep duration; (5) Ideal Health: highest PA and healthy diet and high sleep duration, and the lowest sedentary screen time; and (6) Inactive Healthy Eaters: high healthy diet (Figure 1). The six-cluster solution did not show significant differences related to BMI (F = 1.076, *p* = 0.375). Chi-square showed a non-significant cluster assignment by sex (χ2 = 7.474, df = 5, *p* = >.05). Consequently, sex was not considered a covariate in the following analyses.

Statistically significant differences related to specific types of sedentary screen time were found across clusters (Wilks’ Lambda = 0.606; F (4,441) = 20.000; *p* < 0.001; η_p_^2^ = 0.118). Although all the cluster profiles reported high minutes of different types of sedentary screen time, the Technological Sleepyheads profile reported statistically significant higher minutes than the other profiles. Screen time recommendations in each screen-based device showed a significant association with the six-cluster solution. The Technological Sleepyhead profile showed inhibitory relationships with the different types of sedentary screen time (e.g., ar = −3.7, 12 % meeting TV guidelines). Significant associations between the retained six-cluster solution and adherence to the 24-hour movement recommendations (i.e., PA, screen time, and sleep duration) were found with a medium effect size (χ^2^ = 83.680, df = 15, V = 402, *p* <. 001). Analysis of adjusted residuals revealed that the lowest excitatory relationship (i.e., compliance with more than 2 recommendations) was obtained by the Ideal Health profile and the Active profile, respectively, whereas the highest inhibitory relationship (i.e., compliance with up to 1 recommendation) was obtained by the Ideal Health profile (Table 4).

## 4. Discussion

The main findings of the study revealed that (1) very few Spanish adolescents met the combination of PA, screen time, and sleep duration recommendations, and (2) most of the identified clusters were not exclusively comprised of health-promoting or health-risk behaviors, and were not associated with BMI and sex.

Only a small proportion of adolescents (1.7%) met the overall recommendations (i.e., PA, screen time, and sleep duration) [6], showing lower percentages than previous studies in children [44,45] and adolescents [45,46]. For example, in a recent study conducted in a sample of 1925 children and adolescents, the 80.4% and 53.8% of children boys and girls, respectively, and the 67.5% and 53.8% of adolescent boys and girls, respectively, met PA guidelines [47]. To our knowledge, there are no studies in Spain that have exclusively analyzed the proportion of adolescents meeting all the recommendations of the 24-hour movement guidelines (i.e., PA, screen time, and sleep duration). However, in line with our results, a recent study conducted on Spanish adolescents showed that only 1.2% of Spanish adolescents met the five health behavior recommendations measured in that study (i.e., weight status, screen time, breakfast, PA, and sleep duration) [48]. These results suggest the importance of addressing multiple health behaviors simultaneously, considering an integrative approach to increase PA levels and sleep duration, and decrease sedentary screen time [49]. Given schools provide a lot of opportunities that may be used to promote multiple health-related behaviors among adolescents, it is considered an ideal setting to conduct multiple health behavior change interventions [49,50]. School may also be an effective way to involve teachers, principals, and families in the development of school-based health interventions [51]. For example, incorporating classroom-based PA breaks [52,53], encouraging active commuting to and from school [54], parental restrictions on media use (particularly before and during bedtime) [55], and not allowing sleeping with electronic devices are some of the strategies [56] that could be used to reallocate sedentary time to physical active behaviors or optimal sleep duration.

Our results showed that the proportion of students meeting sleep duration guidelines (89%) was considerably higher than in previous studies [7]. The existence of parental rules, oriented at increasing, or at least, at placing importance on adolescents’ bedtime in this age group, could explain why all the adolescents in the different retained profiles met the sleep duration recommendations [57]. The number of students meeting guidelines for PA (21.4%) and screen time (1.7%) was lower than reported by previous international studies [7,46,58]. However, these percentages are in line with “Spain’s 2018 Report Card on PA for children and youth” [44]. The number of screen-based sedentary behaviors introduced to calculate the total screen time, above all the inclusion of the mobile phone, could explain these low percentages. Regarding healthy diet and ST, our results are slightly better than other studies conducted in Spanish adolescents [59,60].

The inclusion of five health-related behaviors (i.e., PA, ST, healthy diet, sedentary screen time, and sleep duration) revealed six distinct profiles in this study. Overall, none of the identified clusters or subgroups of students were exclusively comprised of healthy or unhealthy behaviors, except for the Ideal Health profile. In line with our results, two of the four retained profiles identified by Dumuid et al. [19,20,21] in children and Sánchez-Oliva et al. [11] in older children, younger adolescents, and older adolescents, were also characterized by healthy and unhealthy behaviors. This clustering of behaviors revealed that healthy or unhealthy behaviors do not necessary ensure that other behaviors go in the same direction [58,61]. In other words, adolescents with high PA levels or low ST seem not to be more likely to comply with other healthy and unhealthy behaviors, respectively. These results are congruent with several previous systematic reviews that revealed a weak association between PA and ST [62,63] or no association between PA and sleep duration in adolescents [12], suggesting that health-related behaviors can co-occur simultaneously [10,18]. Consequently, improving multiple health-related behaviors and simultaneously reducing multiple health risk behaviors becomes of paramount importance for maximizing health benefits in adolescents.

The lack of association between cluster membership and sex are not in line with a previous systematic review, which examined the clustering of PA, ST, and diet in adolescents [9]. A possible explanation for our results may be related to the inclusion of other health-related behaviors in the cluster analysis (i.e., sleep duration and different types of sedentary scree time). Consistent with other international [9] and Spanish studies [64], no significant differences in BMI were found among the six retained clusters. Results are not in line with a recent hypothesis that suggested that meeting recommendations for 24-hour movement guidelines is associated with more appropriate levels of adiposity [5]. One possible justification could be that most of the adolescents in this study met the sleep duration recommendations, which is one risk factor that has been linked to overweight and obesity in adolescents [12,13]. Sleep duration could outweigh unhealthy behaviors, resulting in no association between cluster membership and weight status [9], although further research is needed.

Although all the resulting profiles indicated more than two hours of total screen time, no differences were found in the four types of sedentary screen time (i.e., TV, computer, video games, and mobile phone) in the cluster membership except for the Technological Sleepyhead profile. Contrary to our expectations based on previous studies [29], neither the Active profile nor the Ideal Health profile reported less screen time than other profiles (except for the Technological Sleepyheads profile). This fact could be due to the increase in time spent on screen-based sedentary behaviors such as the use of TVs, computers, videogames, and especially mobile phones in adolescents from developed countries [65]. Recent reports indicated that 99% of adolescents in Spain had a mobile phone [66], which could explain why its use is not associated with healthy behavior. Considering the breakthrough of new technologies in developed countries, practitioners and policy-makers should reconsider the development of new sedentary screen time recommendations for each screen-based sedentary behavior [67].

The Ideal Health and the Active profiles were positively associated with the 24-hour movement guidelines. Given that the combination of high PA levels, low ST levels, and optimal sleep duration, or at least a combination of two of them, are associated with better levels of adiposity, fitness, and cardiometabolic health in youth [5], these two profiles could be considered as more desirable. The results of this study also suggest that meeting one recommendation may not necessarily have a ripple effect on other specific health-related behaviors [58]. On the contrary, the most maladaptive profile was the Inactive Unhealthy Eaters profile, showing no association with meeting 24-hour movement guidelines. The great amount of time spent on different types of screen time in this profile could displace the required number of hours of sleep and PA [11,62]. This holistic approach is grounded in the idea that the whole day matters and should be considered to develop strategies that can lead to establishing an organized schedule that allows adolescents to reduce the time spent on sedentary screen time and increase the time spent on the other movement behaviors (i.e., PA and sleep duration).

### Limitations and Future Directions

This study has several limitations. Firstly, the use of self-reported measures to assess some health-related behaviors such as screen-based sedentary behaviors, sleep duration, diet, and BMI could underestimate or overestimate the results. Secondly, given that the sample was not representative and chosen by convenience and, therefore, some bias could be introduced, results should be interpreted with caution. Thirdly, we only used four (i.e., TV, computer, video games, and mobile phone) of the twelve sedentary behaviors of the Spanish version of Youth Leisure-time Sedentary Behavior Questionnaire (YLSBQ) [36] to avoid completing the questionnaire over a long time period. Therefore, the questionnaire was not adjusted for each behavior as in the original validation. Future studies should use a larger representative sample of Spanish adolescents, with a longitudinal design, to examine the stability of these profiles and their associations with sociodemographic factors and other health-related outcomes, such as tobacco and alcohol consumption.

## 5. Conclusions

This study has revealed that most of the Spanish adolescents failed to meet the combination of PA, sedentary screen time, and sleep duration guidelines. Most of the identified clusters were not exclusively comprised of healthy and unhealthy patterns. These findings reinforce the idea that meeting a particular health recommendation is not likely to have an additional downstream effect on meeting recommendations for other health behavior(s). A lack of association between cluster membership and weight status was found. This could be explained by sleep duration mitigating the combination effect of other healthy and unhealthy behaviors. The equal distribution of sedentary screen time in the different profiles suggests that screen-based sedentary behaviors have become an increasingly important activity in adolescents’ lives. Association of the Ideal Health profile and the Active profile with adherence to the 24-hour movement recommendations reinforces the idea of an optimal time-use balance being important.

## Figures and Tables

**Figure 1 ijerph-16-03151-f001:**
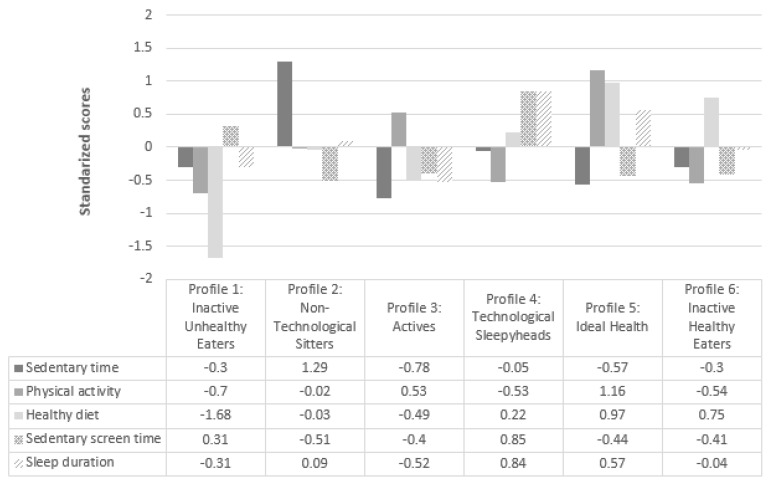
Six-cluster solution based on z-scores for ST, MVPA, healthy diet, sedentary screen time and sleep duration.

**Table 1 ijerph-16-03151-t001:** Sociodemographic characteristics and prevalence of health-related behaviors by sex.

	Total (*n* = 173)	Boys (*n* = 79)	Girls (*n* = 94)	F-value/η_p_^2^
**Age (years; M ± SD)**	12.9 ± 0.5	13.0 ± 0.4	12.9 ±. 58.0	0.019/0.000
**Height (cm)**	161.1 ± 8.3	162.1 ± 9.2	160.2 ± 7.4	2.089/0.012
**Weight (kg)**	49.7 ± 9.8	51.6 ± 10.6	48.2 ± 8.8	5.190 **/0.029
**BMI (kg/m^2^)**	19.0 ± 2.8	19.5 ± 3.0	18.7 ± 2.5	3.698/0.021
**SES (Score: 0–9)**	6.7 ± 1.6	6.6 ± 1.6	6.8 ± 1.6	0.625/0.004
**Physical activity**				
**Daily MVPA levels (M ± SD)**	45.8 ± 15.4	50.3 ± 15.3	41.9 ± 14.4	13.837 **/0.075
**Weekday MVPA (M ± SD)**	50.8 ± 17.8	56.5 ± 19.2	46.1 ± 15.1	15.853 **/0.085
**Weekend day MVPA (M ± SD)**	30.6 ± 23.6	32.6 ± 22.8	29.0 ± 24.3	0.985/0.006
**Sedentary time**				
**Daily ST (M ± SD)**	526.6 ± 51.7	526.5 ± 54.2	526.8 ± 49.7	0.002/0.000
**Weekday ST (M ± SD)**	540.0 ± 57.9	536.4 ± 60.8	543.0 ± 55.5	0.545/0.003
**Weekend day ST (M ± SD)**	489.5 ± 69.3	496.2 ± 67.6	483.8 ± 70.6	1.387/0.008
**Sedentary screen time**				
**Total daily screen time (M ± SD)**	368.8 ± 151.2	385.5 ± 152.9	354.7 ± 149.1	1.785/0.010
**Total weekday screen time (M ± SD)**	325.4 ± 147.4	339.7 ± 150.1	313.3 ± 144.8	1.738/0.008
**Total weekend day screen time (M ± SD)**	477.4 ± 170.5	500.0 ± 172.4	458.3 ± 167.4	2.592/0.015
**Daily TV viewing (M ± SD)**	120.6 ± 65.4	115.8 ± 64.1	124.7 ± 66.6	0.786/0.005
**Daily video game playing (M ± SD)**	68.0 ± 71.5	95.1 ± 77.2	45.2 ± 57.6	23.651 **/0.122
**Daily computer use (M ± SD)**	69.6 ± 66.4	66.5 ± 64.8	72.3 ± 67.9	0.324/0.002
**Daily mobile phone (M ± SD)**	110.4 ± 66.3	108.0 ± 66.5	112.5 ± 66.4	0.197/0.001
**Sleep duration**				
**Daily sleep duration (M ± SD)**	535.7 ± 39.8	533.2 ± 43.7	537.7 ± 36.3	0.542/0.003
**Weekday sleep duration (M ± SD)**	515.2 ± 41.7	517.0 ± 46.3	513.6 ± 37.4	0.286/0.002
**Weekend sleep duration (M ± SD)**	588.4 ± 75.1	575.3 ± 81.8	599.3 ± 67.6	4.455 **/0.025
**Healthy food**				
**Healthy food (Score: 0–6)**	4.5 ± 1.0	4.5 ± 1.0	4.5 ± 0.9	0.016/0.000

Note: M = Medium; SD = Standard deviation; BMI = Body Mass Index; SES = Socioeconomic status; MVPA = Moderate to vigorous physical activity; ST = Sedentary time; TV = Television; * = *p* < 0.05; ** = *p* < 0.01.

**Table 2 ijerph-16-03151-t002:** Compliance with MVPA, screen time, and sleep duration recommendations and combinations of these recommendations by sex.

	Total (*n* = 173)	Boys (*n* = 79)	Girls (*n* = 94)	*x2(df)*/V
**Physical activity**				
**Meeting MVPA recommendations (%)**	37 (21.4%)	25 (31.6%)	12 (12.8%)	9.100(1) **/0.229
**Sedentary screen time**				
**Meeting TV guidelines** **(≤2 h per day) (%)**	79 (45.7%)	39 (49.4%)	40 (42.6%)	0.803(1)/0.068
**Meeting video games guidelines** **(≤2 h per day) (%)**	132 (76.3%)	51 (64.4%)	81 (86.2%)	11.809(1) **/0.253
**Meeting computer guidelines** **(≤2 h per day) (%)**	139 (80.3%)	64 (81%)	75 (79.8%)	0.041(1)/0.015
**Meeting mobile phone guidelines** **(≤2 h per day) (%)**	101 (58.4%)	48 (60.8%)	53 (56.4%)	0.338(1)/0.044
**Meeting all total screen time recommendations (%)**	3 (1.7%)	2 (2.5%)	1 (1.1%)	0.543(1)/0.056
**Sleep duration**				
**Meeting sleep duration recommendations (%)**	154 (89%)	67 (84.8%)	87 (92.6%)	2.632(1)/0.123
**Meeting recommendations for 24-hour movement guidelines (PA, screen time, and sleep duration)**				
**Not meeting recommendations (%)**	15 (8.7%)	9 (11.4%)	6 (6.4%)	9.768(3) **/0.238
**Meeting exclusively one recommendation (%)**	125 (72.3%)	48 (60.8%)	77 (81.9%)	
**Meeting exclusively two recommendations (%)**	30 (17.3%)	20 (25.3%)	10 (10.6%)	
**Meeting three recommendations (%)**	3 (1.7%)	2 (2.5%)	1 (1.1%)	

Note: MVPA = Moderate to vigorous physical activity; TV = Television; * = *p* < 0.05; ** = *p* < 0.01.

**Table 3 ijerph-16-03151-t003:** Health-related behaviors for the six-cluster solution.

Variables	Cluster 1: Inactive Unhealthy Eaters (*n* = 20; 11.56%)	Cluster 2: Non-Technological Sitters (*n* = 41; 23.69%)	Cluster 3: Active (*n* = 25; 14.45%)	Cluster 4: Technological Sleepyheads (*n* = 25; 14.45%)	Cluster 5: Ideal Health (*n* = 23; 13.29%)	Cluster 6: Inactive Healthy Eaters (*n* = 39; 22.54%)	F-value	η_p_^2^
**Daily ST levels (M ± SD)**	522.8 ^a,b^ (39.0)	591.4 ^a,c,d,e,f^ (35.7)	484.2 ^b,c,g^ (29.6)	522.0 ^d,g^ (35.7)	495.1 ^e^ (33.8)	509.3 ^f^ (36.6)	34.025 **	0.50
**Daily MVPA levels (M ± SD)**	34.4 ^a,b,c^ (11.1)	46.34 ^a,d,e,f,g^ (12.3)	56.2 ^b,d,h,i,j^ (10.8)	37.3 ^e,h,k^ (12.7)	67.2 ^c,f,i,k,l^ (8.6)	37.2 ^g,j,l^ (9.4)	40.504 **	0.54
**Healthy diet (Score: 0-6)**	2.8 ^a,b,c,d,e^ (0.4)	4.4 ^a,f,g^ (0.6)	4.00 ^b,h,i,j^ (0.6)	4.7 ^c,h,k,l^ (0.7)	5.4 ^d,f,i,k^ (0.5)	5.2 ^e,g,j,l^ (0.5)	57.295 **	0.63
**Daily sedentary screen time (M ± SD)**	458.7 ^a,b,c,d^ (181.1)	306.2 ^a,e^ (100.5)	326.3 ^b,f^ (145.9)	557.7 ^e,f,g,h^ (103.5)	318.8 ^c,g^ (134.4)	324.1 ^d,h^ (95.2)	28.338 **	0.45
**Daily sleep duration** **(M ± SD)**	515.6 ^a,b^ (40.9)	536.6 ^c,d^ (38.2)	505.4 ^c,e,f^ (36.1)	571.5 ^a,d,e,g^ (28.8)	558.9 ^b,f,h^ (34.2)	528.9 ^g,h^ (26.7)	23.128 **	0.38

Note: Values in parentheses are standard deviation. A cluster mean is significantly different to another mean if it has the same superscripts. * = *p <* 0.05; ** = *p <* 0.01.

**Table 4 ijerph-16-03151-t004:** Association between resulting clusters with recommendations for the 24-hour movement guidelines.

Recommendations	0	1	2	3
	n (%) ar	n (%) ar	n (%) ar	n (%) ar
**Cluster 1:** **Inactive Unhealthy Eaters**	5 (25%) 2.8	15 (75%) 0.3	0 (0%) −2.2	0 (0%) −0.6
**Cluster 2:** **Non-technological Sitter**	2 (4.9%) −1.0	30 (73.2%) 0.2	9 (22%) 0.9	0 (0%) −1.0
**Cluster 3:** **Active**	3 (12%) 0.6	15 (60%) −1.5	5 (20%) 0.4	2 (8%) 2.6
**Cluster 4:** **Technological Sleepyhead**	4 (16%) 1.4	21 (84%) 1.4	0 (0%) −2.5	0 (0%) −0.7
**Cluster 5:** **Ideal Health**	0 (0%) −1.6	6 (26.1%) −5.3	16 (69.6%) 7.1	1 (4.3%) 0
**Cluster 6:** **Inactive Healthy Eaters**	1 (2.6%) −1.5	38 (97.4%) 4.0	0 (0%) −3.3	0 (0%) −0.9

Note: *n* = subject frequency; % = percentage; ar = adjusted residuals.

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
