# Peer review of "Healthy or Unhealthy? The Cocktail of Health-Related Behavior Profiles in Spanish Adolescents"

_ijerph, 2019, doi:10.3390/ijerph16173151_

Round 1

Reviewer 1 Report

I appreciate the opportunity to review this interesting and newfangled paper entitle "Healthy or unhealthy? The “cocktail” of health-related behavior profiles in Spanish adolescents". In this work the authors identified the prevalence and clustering of health-related behaviors in Spanish adolescents and to examine their association with sex, body mass index, different types of sedentary screen time, and adherence to the 24-hour movement guidelines.

It is a study of good quality that I think will be relevant for the readers of the IJERPH. Clearly a lot of work went into its construction. However, there are some comments that have to be taken into account. Below are my comments for the authors:

Page 1, lines 28 and 29: Please      avoid keywords that have been previously included in the title.

Page 1, line 32: The reference      number 2 showed the inverse association between the time spent in      sedentary behaviors, such as sedentary screen time, and health outcomes in      youth. However, recent studies have questioned the relationship between      device-measured sedentary time and health outcomes in youth. The authors      should justify the use of accelerometer-measured sedentary time in this      study.

Page 2, lines 43-45: The      authors omitted an important paper about lifestyle clusters in Spanish      children and adolescents: Sánchez-Oliva et al. (2018). J Pediatr, 230, 317-324. It should      be use at least in the Discussion section.

Page 3, lines 106-111: What      criteria was followed to decide the sample size? Is it enough for the design      and purpose of this study?

Page 3, line 118: Why did you      choose the WHO obesity criteria instead of the IOTF or CDC criteria? Could      this decision influence on your results?

Page 4, lines 138-149: The      YLSBQ should be adjusted for improve their psychometric properties (see      the reference number 35). However, all its items are necessary to apply      the adjustment syntax. How did the authors proceed in this regard?

Page 5, lines 186-210: Please      be careful with the aesthetics of the manuscript.

Page 6, Table 1: Please show      the decimal when it is zero (for example, 13±0.4 vs. 68.0±71.5).

Pages 6 and 7, Tables 1 and 2:      Tables should be self-explained. Please show their abbreviations in notes.

Page 10, lines 296-298: Are the      school-based interventions more effectives to promote multiple      health-related behaviors in adolescents than interventions based in other      settings?

Page 10, lines 304-305: For      information, recent studies in a large sample of Spanish youth also      include the percentage of them meeting guidelines for PA (Grao-Cruces et      al. (2019) J Sch Health, 89(8),      612-618; Grao-Cruces et al. (2019) Scand      J Med Sci Sports, 29(4), 554-565).

Page 10, line 308: Please check      the use of the references number 51 and 52.

Pages 10 and 11, lines 326-345:      One of the main limitations of this work is the use of self-reported      measures. It would be good to provide information in the Methods section about      the correlation between objectively measure BMI and self-reported BMI in      adolescents.

Author Response

Comments and Suggestions for Authors

I appreciate the opportunity to review this interesting and newfangled paper entitle "Healthy or unhealthy? The “cocktail” of health-related behavior profiles in Spanish adolescents". In this work the authors identified the prevalence and clustering of health-related behaviors in Spanish adolescents and to examine their association with sex, body mass index, different types of sedentary screen time, and adherence to the 24-hour movement guidelines.

It is a study of good quality that I think will be relevant for the readers of the IJERPH. Clearly a lot of work went into its construction. However, there are some comments that have to be taken into account. Below are my comments for the authors:

Answer: We truly appreciate your kind words regarding the efforts made to undertake this study. Likewise, we would like to thank you for the quality of the comments made in your review to try to improve our manuscript. Below, you will find the answers to each specific comment that you have developed in the review.

Page 1, lines 28 and 29: Please avoid keywords that have been previously included in the title.

Answer: The suggested change has been made. The term “adolescents” has been removed from the keywords.

Page 1, line 32: Reference number 2 showed the inverse association between the time spent in sedentary behaviors, such as sedentary screen time, and health outcomes in youth. However, recent studies have questioned the relationship between device-measured sedentary time and health outcomes in youth. The authors should justify the use of accelerometer-measured sedentary time in this study.

Answer: We agree with the reviewer that while some screen time behaviors (e.g., TV, video games, computer, etc.) has been negatively related to health indicators, inconsistent findings have been found for other screen time behaviors (e.g., doing homework and reading) in school-aged children and youth (Carson et al., 2016). However, we agree that this relation is less apparent when sedentary time is measured with objective movement counters or position monitors. For example, studies using objective accelerometer measures of sedentary time yielded null associations with adiposity in youth (Biddle, García, & Wiesner, 2017). However, it also should be noted that there is reasonable evidence for a likely causal relationship between sedentary time and all-cause mortality (Biddle et al., 2016). Therefore, we agree with the reviewer that current evidence suggests screen time has a bigger impact on health compared with overall sedentary time. Therefore, considering the aforementioned arguments and the reviewer's suggestion, we have changed the term “sedentary time” by “sedentary screen time” in the first paragraph and we also clarified in this section the screen time behaviors that are negatively related to health indicators.

Given that ST and sedentary screen time are associated differently with different health indicators (Biddle et al., 2017; Carson et al., 2016), their clustering should be independently studied. This information is already available in the manuscript (Page 3, Line 87-88). Moreover, in line with previous cluster studies (Dumuid et al., 2017; Sánchez-Oliva et al., 2018) some authors suggest the importance of combining objective measures with subjective measures to capture more specific information (Carson et at., 2016).

Page 2, lines 43-45: The authors omitted an important paper about lifestyle clusters in Spanish children and adolescents: Sánchez-Oliva et al. (2018). J Pediatr, 230, 317-324. It should be used at least in the Discussion section.

Answer: Considering the reviewer's suggestions, the suggested reference has been added both in the introduction and discussion section.

Page 3, lines 106-111: What criteria was followed to decide the sample size? Is it enough for the design and purpose of this study?

Answer: To our knowledge, there are no well-defined rules-of-thumb about the sample size necessary to do cluster analysis. Due to the lack of rules, one of the recommendations that can be given concerning sample sizes and variable numbers is to critically question whether the dimensionality is not too high for grouping the number of cases (Dolnicar, 2002). Drawing on “classic” rules of thumb, Formann (1984) suggests the minimal sample size to include no less than 2k cases (k = number of variables), preferably 5*2k. In our study, we followed the recommendations of Formann (1984) = 5*2k, wherek” is the number of clustering variables for algorithmic approaches. Applying the Formann formula, the minimum sample size recommendation was met for this study (the minimum number of students had to be at least 160).

As one can see in most of the previous cluster studies on social and health sciences, no rules-of-thumb about the sample size necessary for cluster analysis are considered (perhaps due to the nature of the studies). For instance, Van den Berghe et al. (2013) conducted a cluster analysis with 93 physical education teachers and two variables (i.e., autonomous and controlled motivation). In addition, Dumuid et al. (2017) examined the clustering of the following lifestyle behaviors as cluster inputs: light, moderate, and vigorous physical activity, sedentary behavior, sleep duration, screen time, and diet (7 variables) in 285 Australian children aged 9-11 years.

Nevertheless, despite there not being any general recommendations in the field of social and health sciences to ensure that the available sample size is sufficient for the cluster analysis, we acknowledge in the limitations section that the sample was not representative and caution is, therefore, recommended for generalizing results.

Page 3, line 118: Why did you choose the WHO obesity criteria instead of the IOTF or CDC criteria? Could this decision influence on your results?

Answer: Some studies that have compared the World Health Organisation (WHO) and the International Obesity Task Force (IOTF) Body Mass Index cut-offs in estimating overweight or obesity in children and adolescents have suggested that given the seriousness of the obesity epidemic, the WHO system should be preferable to the national standards for clinical practice and/or obesity screening (Valerio et al., 2017). The WHO standards have also been well received worldwide and, at the time of this writing, they have been adopted by over 110 countries and many researchers (De Onis & Lobstein, 2010). Some previous studies that calculated the self-reported BMI have only used the World Health Organization reference data (de Onis et al., 2007). (Dumuid et al., 2017; Dumuid et al., 2018). However, if the reviewer considers it necessary, we could report this fact as a limitation.

Page 4, lines 138-149: The YLSBQ should be adjusted for improve their psychometric properties (see the reference number 35). However, all its items are necessary to apply the adjustment syntax. How did the authors proceed in this regard?

Answer: In the present study, we only used four (i.e., TV, computer, video games, and mobile phone) of the twelve sedentary behaviors of the Spanish version of Youth Leisure-time Sedentary Behavior Questionnaire (YLSBQ). Therefore, we do not apply the adjustment syntax. This information has been provided in this instrument to avoid misunderstandings and in the “limitations” section as one of the limitations of this study.

Page 5, lines 186-210: Please be careful with the aesthetics of the manuscript.

Answer: The suggested change has been made.

Page 6, Table 1: Please show the decimal when it is zero (for example, 13±0.4 vs. 68.0±71.5).

Answer: The suggested changes have been made.

Pages 6 and 7, Tables 1 and 2: Tables should be self-explained. Please show their abbreviations in notes.

Answer: The suggested changes have been made.

Page 10, lines 296-298: Are the school-based interventions more effectives to promote multiple health-related behaviors in adolescents than interventions based in other settings?

Answer: To our knowledge, there is not enough evidence yet to support this claim. We just mentioned that school provides a lot of opportunities that may be used to promote multiple health-related behaviors among adolescents. This sentence has been re-written in order to further clarify its comprehension for the reader. Additional examples have been added to support this assumption.

Page 10, lines 304-305: For information, recent studies in a large sample of Spanish youth also include the percentage of them meeting guidelines for PA (Grao-Cruces et al. (2019) J Sch Health, 89(8), 612-618; Grao-Cruces et al. (2019) Scand J Med Sci Sports, 29(4), 554-565).

Answer: Considering the reviewer's suggestions, the suggested references have been added in the discussion section.

Page 10, line 308: Please check the use of the references number 51 and 52.

Answer: We sincerely apologize for this issue because it was a mistake in the writing process of the manuscript.

Pages 10 and 11, lines 326-345: One of the main limitations of this work is the use of self-reported measures. It would be good to provide information in the Methods section about the correlation between objectively measure BMI and self-reported BMI in adolescents.

Answer: We agree with the reviewer that self-reported weight and height is a source of error. As we acknowledged in the limitations of the study, the use of some self-reported measures such as BMI could underestimate or overestimate the results. However, while some studies showed that BMI is a poor predictor of adiposity in young overweight and obese children (He, Cai, & Fan, 2018), other studies showed that BMI might be a viable alternative when direct measurement of BMI is not available as in our study (Fonseca et al., 2010; Pérez, Gabriel, Nehme, Mandell, & Hoelscher, 2015). Given that in this study BMI could not be considered as one of the main study variables, we really believe that this method of measurement does not represent a significant source of error in this design, although it is a limitation of this study.

References

Biddle, S. J., Bengoechea, E. G., & Wiesner, G. (2017). Sedentary behaviour and adiposity in youth: a systematic review of reviews and analysis of causality. International Journal of Behavioral Nutrition and Physical Activity14(1), 43.

Biddle, S. J., Bennie, J. A., Bauman, A. E., Chau, J. Y., Dunstan, D., Owen, N., ... & van Uffelen, J. G. (2016). Too much sitting and all-cause mortality: is there a causal link? BMC Public Health16(1), 635.

Carson, V., Hunter, S., Kuzik, N., Gray, C. E., Poitras, V. J., Chaput, J. P., ... & Kho, M. E. (2016). Systematic review of sedentary behaviour and health indicators in school-aged children and youth: an update. Applied Physiology, Nutrition, and Metabolism41(6), 240-265.

De Onis, M., & Lobstein, T. (2010). Defining obesity risk status in the general childhood population: which cut-offs should we use? International Journal of Pediatric Obesity5(6), 458-460.

Dolnicar, S., Grün, B., Leisch, F., & Schmidt, K. (2014). Required sample sizes for data-driven market segmentation analyses in tourism. Journal of Travel Research, 53(3), 296-306.

Dumuid, D., Olds, T., Martín-Fernández, J. A., Lewis, L. K., Cassidy, L., & Maher, C. (2017). Academic performance and lifestyle behaviors in Australian school children: A cluster analysis. Health Education & Behavior44(6), 918-927.

Dumuid, D., Stanford, T. E., Pedišić, Ž., Maher, C., Lewis, L. K., Martín-Fernández, J. A., ... & Tremblay, M. S. (2018). Adiposity and the isotemporal substitution of physical activity, sedentary time and sleep among school-aged children: A compositional data analysis approach. BMC Public Health, 18(1), 311.

Fonseca, H., Silva, A. M., Matos, M. G., Esteves, I., Costa, P., Guerra, A., & Gomes‐Pedro, J. (2010). Validity of BMI based on self‐reported weight and height in adolescents. Acta Paediatrica99(1), 83-88.

Formann, A. K. (1984). Die Latent-Class-Analyse: Einfuhrung in die Theorie und Anwendung [Latent class analysis: Introduction to theory and application]. Weinheim: Beltz.

He, J., Cai, Z., & Fan, X. (2018). How accurate is the prevalence of overweight and obesity in children and adolescents derived from self-reported data? A meta-analysis. Public Health Nutrition21(10), 1865-1873.

Pérez, A., Gabriel, K. P., Nehme, E. K., Mandell, D. J., & Hoelscher, D. M. (2015). Measuring the bias, precision, accuracy, and validity of self-reported height and weight in assessing overweight and obesity status among adolescents using a surveillance system. International Journal of Behavioral Nutrition and Physical Activity12(1), 2.

Sánchez-Oliva, D., Grao-Cruces, A., Carbonell-Baeza, A., Cabanas-Sánchez, V., Veiga, O. L., & Castro-Piñero, J. (2018). Lifestyle Clusters in School-Aged Youth and Longitudinal Associations with Fatness: The UP&DOWN Study. The Journal of Pediatrics203, 317-324.

Valerio, G., Balsamo, A., Baroni, M. G., Brufani, C., Forziato, C., Grugni, G., ... & Pacifico, L. (2017). Childhood obesity classification systems and cardiometabolic risk factors: a comparison of the Italian, World Health Organization and International Obesity Task Force references. Italian journal of pediatrics43(1), 19.

Van den Berghe, L., Cardon, G., Aelterman, N., Tallir, I. B., Vansteenkiste, M., & Haerens, L. (2013). Emotional exhaustion and motivation in physical education teachers: A variable-centered and person-centered approach. Journal of Teaching in Physical Education32(3), 305-320.

Reviewer 2 Report

Add a paragraph in the "Data analysis" section, for example, where you specify the recommended values ​​for each of the behaviors. Because, although this information is referred to in the introduction in a dispersed way for each behavior, as it is a data in the analysis of the results, it was important to mention in this section what was considered recommended for each behavior.

Author Response

Comments and Suggestions for Authors

Add a paragraph in the "Data analysis" section, for example, where you specify the recommended values ​​for each of the behaviors. Because, although this information is referred to in the introduction in a dispersed way for each behavior, as it is a data in the analysis of the results, it was important to mention in this section what was considered recommended for each behavior.

Answer: We would like to thank you for the quality of your comment made in your review to improve our manuscript. We agree with the reviewer that this information was not clear enough. To facilitate readers’ comprehension of the study and taking the reviewer's suggestion into account, the physical activity, sedentary screen time, and sleep duration recommendations have been added (Page 5, Line 169-172). We hope this change will help readers to better understand the most important results of our manuscript.